# Chinese herbal medicine for post-viral fatigue: A systematic review of randomized controlled trials

Le-Yan Hu[1], An-Qi Cai[1], Bo Li[1], Zheng Li[2], Jian-Ping Liu[1], Hui-Juan Cao[1]*

1 Beijing University of Chinese Medicine, Beijing, China, 2 Traditional Chinese Medicine Hospital affiliated to Xinjiang Medical University, Ürümqi, Xinjiang, China

* huijuancao327@hotmail.com

## Abstract

### Background

Fatigue is a common symptom after viral infection. Chinese herbal medicine (CHM) is thought to be a potential effective intervention in relieving fatigue.

### Purpose

To assess the effectiveness and safety of CHM for the treatment of post-viral fatigue.

### Study design

Systematic review and meta-analysis of randomized controlled trials (RCTs).

### Methods

The protocol of this systematic review was registered on PROSPERO (CRD42022380356). Trials reported changes of fatigue symptom, which compared CHM to no treatment, placebo or drugs, were included. Six electronic databases and three clinical trial registration platforms were searched from inception to November 2023. Literature screening, data extraction, and risk bias assessment were independently carried out by two reviewers. Quality of the included trials was evaluated using Cochrane risk of bias tool, and the certainty of the evidence was evaluated using GRADE. The meta-analysis was performed using Review Manager 5.4, mean difference (MD) and its 95% confidence interval (CI) was used for estimate effect of continuous data. Heterogeneity among trials was assessed through $I^2$ value.

### Results

Overall, nineteen studies with 1921 patients were included. Results of individual trial or meta-analysis showed that CHM was better than no treatment (MD = -0.80 scores, 95%CI -1.43 to -0.17 scores, $P$ = 0.01, 60 participants, 1 trial), placebo (MD = -1.90 scores, 95%CI -2.38 to -1.42 scores, $P<0.00001$, 184 participants, 1 trial), placebo on basis of rehabilitation therapy (MD = -14.90 scores, 95%CI -24.53 to -5.27 scores, $P$ = 0.02, 118 participants, 1 trial) or drugs (MD = -0.38 scores, 95%CI -0.48 to -0.27 scores, $I^2$ = 0%, $P<0.00001$, 498

**Data Availability Statement:** All relevant data are within the manuscript and its Supporting information files.

**Funding:** This work is funded by the key R&D project of Xinjiang Autonomous Region (Research

on key technologies of prevention and control for COVID-19, 2021B03003-2) and the Educational science research project of Beijing University of Chinese Medicine (XBB23071). The funders had no role in study design, data collection and analysis, decision to publish, or preparation of the manuscript.

**Competing interests:** The authors have declared that no competing interests exist.

**Abbreviations:** 95% CI, 95% Confidence Interval; AIDS, Acquired Immune Deficiency Syndrome; BFI, Brief Fatigue Inventory; C, Control group; CFS, chronic fatigue syndrome; CHM, Chinese Herbal Medicine; CLDQ, Chronic Liver Disease Questionnaire; COVID-19, Corona Virus Disease 2019; CRF, Cancer-Related Fatigue; ETV, Entecavir; FS-14, fatigue scale-14; HAART, Highly Active Antiretroviral Therapy; HM, Herbal Medicine; INF, Interferon; LAM, Lamivudine; m, months; MD, mean difference; NR, not report; RBV, Ribavirin; RCT, Randomized Controlled Trials; RR, Risk Ratio; T, Treatment group; TCM, Traditional Chinese Medicine; TDF, Tenofovir Disoproxil Fumarate; Vit, vitamin; VMC, Viral Myocarditis; w, weeks.

participants, 4 trials) on relieving fatigue symptoms assessing by Traditional Chinese Medicine fatigue scores. Trials compared CHM plus drugs *to* drugs alone also showed better effect of combination therapy (average MD = -0.56 scores). In addition, CHM may improve the percentage of CD4 T lymphocytes and reduce the level of serum IL-6 (MD = -14.64 scores, 95%CI 18.36 to -10.91 scores, $I^2$ = 0%, P<0.00001, 146 participants, 2 trials).

## Conclusion

Current systematic review found that the participation of CHM can improve the symptoms of post-viral fatigue and some immune indicators. However, the safety of CHM remains unknown and large sample, high quality multicenter RCTs are still needed in the future.

## Introduction

Fatigue is a subjective uncomfortable feeling symptom, which lacks a unified definition. Patients describe fatigue as a persistent overwhelming feeling of tiredness, severe lethargy, or lack of energy, physical weakness, and cognitive decline [1]. As a common non-specific symptom [2], fatigue can be the main patients' complaint or the accompanying symptom of a variety diseases. Some fatigue can be relieved soon through adequate rest or treatment, while many fatigues cannot be relieved and may last for a long time, causing harm to people's psychology and physiology, seriously interfering with people's life [3].

There are many reasons for fatigue, among which more and more evidence shows that viral infection, such as Epstein-Barr virus, Bernard virus, human herpes virus, cytomegalovirus, etc, is related to fatigue [4]. After infection, some people will leave fatigue symptoms of long time and repeated attacks but the specific mechanism is still unclear. In addition, human immunodeficiency virus (HIV), hepatitis B virus and other infections often cause long-term fatigue symptoms of infected people. The results of AIDS and hepatitis B with large sample study in China show that the prevalence rate is up to 40% [5, 6] and even more than 60% in England and North America [7, 8], which cannot be alleviated after the application of antiviral drugs [9, 10]. In recent years, with the prevalence of COVID-19, people pay more attention to the symptoms and complications after acute viral infection, which are called "Long COVID", including symptoms of long-term fatigue [11]. An meta-analysis have shown that approximately a third of individuals experienced persistent fatigue 12 or more weeks following confirmed COVID-19 diagnosis [12]. The chronic fatigue after the virus infection not only leads to the emergence of anxiety, depression, and serious decline in the quality of life [13], but also increases the economic burden of the whole society [14].

At present, treatment of fatigue is mainly symptomatic. Among them, drug therapy mainly includes antidepressants, vitamin or other nutritional therapies to improve clinical symptoms. Non drug therapy includes cognitive behavioral therapy and exercise therapy to relieve patients' anxiety, depression and other negative emotions [15], but it cannot fundamentally solve the problem and has not achieved satisfactory results [16].

According to the theory of Traditional Chinese Medicine (TCM), the disease mechanism of long-term fatigue is the deficiency in origin and excess in superficiality, due to excessive exertion, improper diet, internal emotional injury or exogenous pathogenic factors (e.g. viral infection) causing the deficiency of *Qi*, blood, *Yin* or *Yang*, then mixing with fire, phlegm and blood stasis [17]. Among them, the healthy *Qi* of the human body is the key factor [18]. TCM therapies, including Chinese herbal medicine (CHM), acupuncture, massage, Qigong and

other non-pharmaceutical treatments, can play a complementary role in the treatment of chronic fatigue [19, 20]. CHM can help tonifying *Qi* and blood, regulating *Yin* and *Yang*, supplying deficiency and expelling excess, which is widely used in China [21].

At present, although there is no direct evidence for the usage rate of CHM in the treatment of post-viral fatigue, but in the past few decades, it has been widely used in clinic and achieved certain results in the treatment of cancer-related fatigue (CRF) and chronic fatigue syndrome (CFS). Evidence suggests that CHM can clearly decreased 1.77 scores in fatigue scale-14 (FS-14) scores as an adjuvant or monotherapy for CFS [22] and significantly lowered 1.47 scores in Brief Fatigue Inventory (BFI) global score for CRF [23]. Meanwhile take Ginseng for example [24], up to now, various clinical practice and animal-based experiments have already confirmed the safe anti-fatigue effects of Ginseng Radix et Rhizoma, as well as its components. Ginseng Radix et Rhizoma is currently prescribed in the formula for Long COVID treatment as a monarch drug. Among these formulas, Qingjin Yiqi granules can significantly alleviate fatigue and have been recommended by the Rehabilitati on Guidelines of Integrated Medicine for Long COVID treatment in clinical.

Currently, there are two systematic reviews published in Pubmed evaluated the efficacy of interventions for post-viral fatigue, the interventions included mixed- methods [25] (e.g. exercise, psychological/behavioral, dietary, acupuncture or yoga) and Vitamin C [26]. There is no evidence of evaluated the efficacy of CHM in the treatment of fatigue after viral infection. Therefore, this review aims to comprehensively search and critical appraisal all relevant randomized controlled trials (RCT) to explore the effectiveness and safety of CHM for the treatment of post-viral fatigue.

## Methods

This review was reported according to PRISMA statement (S1 Appendix). The protocol of this systematic review was registered on PROSPERO, which is available on https://www.crd.york.ac.uk/prospero/display_record.php?ID=CRD42022380356 (CRD42022380356).

### Eligibility criteria

We focused on post-viral fatigue, thus, those fatigue caused by mental illness (e.g. depression, anxiety disorders), cancer, anemia, endocrine disorders or chronic somatic diseases were not included.

We included studies if they met the following criteria:

1. Parallel-group RCTs with or without blinding method. Only the first phase data of randomized cross-over trials were included to avoid any carry-over effects.

2. Participants of any age across all regions and ethnicity, that had mild or above fatigue symptom assessed by validated instruments which had clear fatigue classification as a result of viral infection (The diagnosis is confirmed by pathogenic examination). At the same time, patients were enrolled with clear TCM diagnosis in line with the category of deficiency syndrome (including *Qi* deficiency, blood deficiency, *Yin* deficiency and *Yang* deficiency).

3. The intervention was oral administration of CHM, dosage form, dosage and composition of the prescription were not limited.

4. The control group was no treatment, placebo or anti-fatigue drugs. CHM combined with other treatments compared to other treatments alone were also included.

5. The primary outcome was fatigue symptom evaluation via score or fatigue scale (e.g. Chalder fatigue scale [27]), symptom disappearance rate and disappearance time. The

secondary outcomes included quality of life measured by validated instruments (e.g. chronic liver disease questionnaire, CLDQ [28]), laboratory characteristics on immune-related indicators (e.g. CD4 +, CD8+), pathogen related outcome indicators (e.g.HbsAg, Hbv DNA) and adverse events.

Exclusion criteria:

1. Incomplete data or full text not available

2. Duplicate publication

## Information sources

We searched six electronic databases and three clinical trial registration platforms, including China Network Knowledge Infrastructure (CNKI), China Scientific Journals Database(VIP), Wan Fang Database, Chinese Biomedical Literature Serving System (SinoMed), PubMed, the Cochrane Central Register of Controlled Trials (CENTRAL), the US National Institutes of Health Ongoing Trials Registry, The Chinese Clinical Trial Registry and the World Health Organization International Clinical Trials Registry platform from their inception to 19th November 2023.

## Search strategy

We used the combined method of MeSH Term and free words by applying the following terms: "traditional Chinese medicine", "herbal medicine", "Chinese patent medicine", "post viral fatigue", "post viral fatigue", "chronic fatigue", "fatigue", "chronic fatigue syndrome", "viral", "viruses", "serious acute respiratory syndrome", "SARSCoV", "covid-19", "H1N1", "influenza", "H7N9", "hepatitis a", "hepatitis b", "hepatitis c", "hepatitis d", "hepatitis e", "HIV", "AIDS", "Infectious Mononucleosis Herpes Zoster", "Hand Foot and Mouth Disease", "Encephalitis Japanese", "Cytomegalovirus", "Infectious Mononucleosis", "Hemorrhagic Fever with Renal Syndrome", "Enterovirus", "Dengue", "randomly", "trial", "randomised", "Randomized Controlled Trial" and "randomized". Details of the searching strategy of CNKI and PubMed were shown in S2 Appendix.

## Selection process

Citation of the retrieved literature exported from 6 electronic databases and three clinical trial registration platforms, and was imported into NoteExpress software to preliminary exclude the duplicates. Titles, abstracts and full text screened by two reviewers (Hu LY and Cai AQ) independently according to the eligibility criteria. Disagreements resolved by the third author (Cao HJ).

## Data collection

Two authors (Hu LY and Cai AQ) independently extracted data form all eligible articles into a data extraction form. The conflicts resolved through discussion and consultation with a third author (Cao HJ).

We extracted the following information for each trial:

1. Basic information of research: title, first author, year of publication, journal information

2. Information of study design: methods of the randomization, allocation concealment, blinding, dealing with the missing data, founding sources

3. Characteristics of participants: gender, sample size, mean age, type of disease, syndrome of TCM, severity of fatigue, disease duration

4. Intervention: name of the prescription, components of the herbs and dosage, method of administration, dosage form, frequency and duration of the treatment

5. Control: type of control, details of the control treatment

6. Outcomes: type of the outcomes, data of the results, timing of the measurement, follow-up period, adverse event

## Methodological quality assessment

The quality of the included trials was assessed using the Cochrane Collaboration's "Risk of bias (RoB) 2.0" tool [29] by two authors (Hu LY and Cai AQ). Three types of results were obtained: "low risk," "high risk," and "some concerns." The evaluation included randomization process, deviations from intended interventions, missing outcome data, measurement of the outcome, and selection of the reported result.

## Data analysis

All statistical analysis was performed using Review Manager (Revman 5.4, The Cochrane Collaboration) software. Data was summarized using risk ratio (RR) with its 95% confidence interval (CI) for binary outcomes or mean difference (MD) with 95% (CI) for continuous outcomes. Statistical heterogeneity among included trials was evaluated by the $I^2$ value. Meta-analysis was conducted if there was no significant clinical (in term of difference of participants, intervention, control and outcomes) and statistical heterogeneity ($I^2 < 75\%$) among included trials. Considering the complexity of the herbal formula, random effects model was used in meta-analysis, pooling analysis was not be used if there was significant statistical heterogeneity ($I^2 \geq 75\%$) among trials and the source of heterogeneity could not be interpret by subgroup analysis. The funnel plot was used to explore the possibility of small study effects or publication bias, if there were ten or more studies in an analysis.

## Sensitivity analysis and subgroup analysis

We implemented sensitivity analyses to explore the impacts of evidence quality on the robustness of review conclusions. Subgroups were classified by types of viral infection, patients' age (<18 years, ≥18 years), treatment principle (e.g. benefiting *Qi*, nourishing *Yin*, tonify *Yang)*, level of fatigue symptom, duration of treatment (short-term is defined as less than or equal to 1 month, middle-term is 1–3 months, long-term is 3–6 months).

## Evaluation on the certainty of the evidence

We evaluated the quality of evidence using GRADE (Grading of Recommendation, Assessment, Development, and Evaluation) [30]. Evidence was rated as high, moderate, low, or very low according to study limitations, inconsistency, indirectness, imprecision, and possibility of publication bias.

# Results

A preliminary literature search identified 3470 records, and 307 studies were screened out after deleting duplicates. Finally, we included 19 trials through full-text screening. The PRISMA 2020 flow diagram for literature screening is shown in the Fig 1.

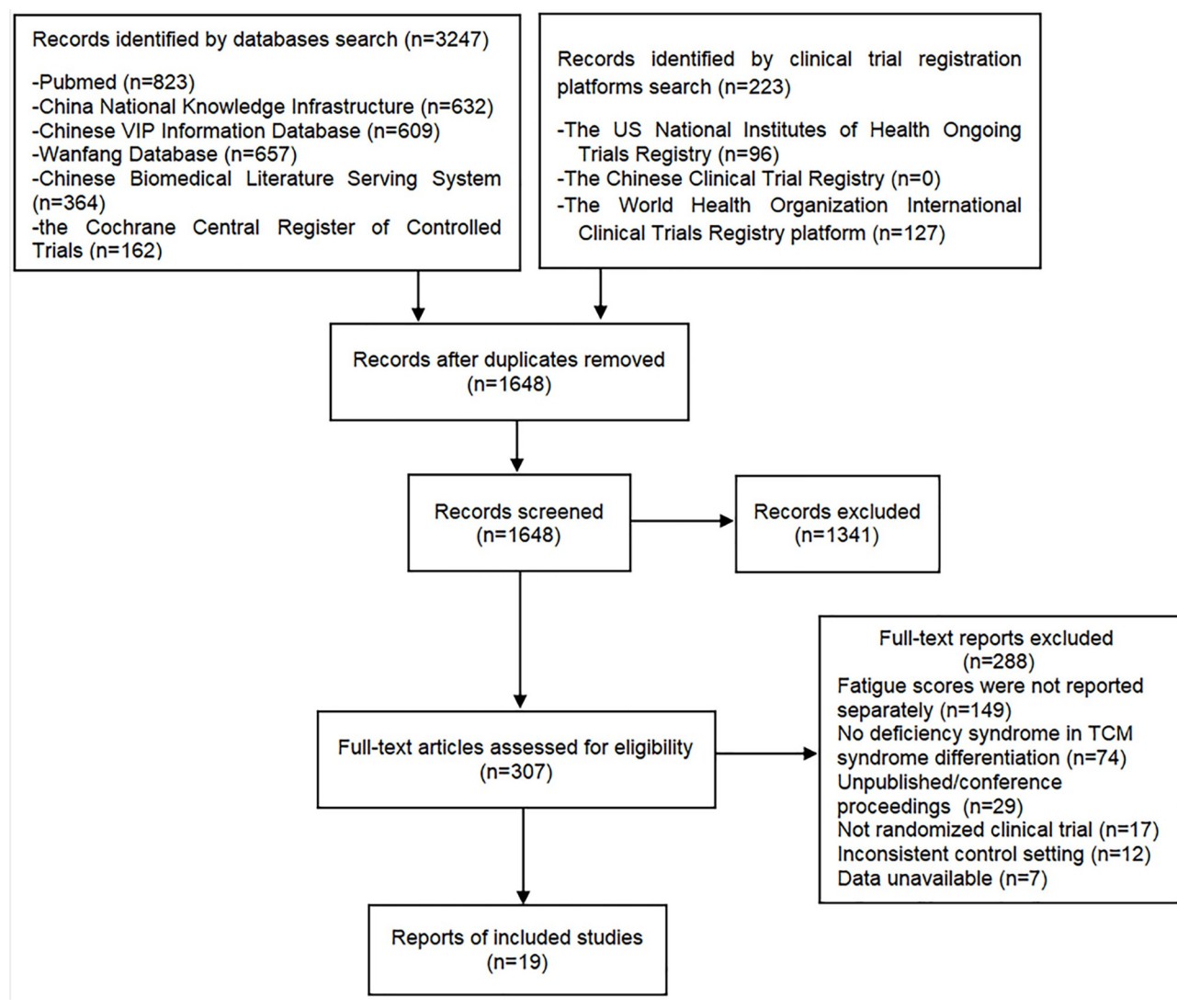

**Fig 1. Flow chart of study selection.**

## Study characteristics

A total of 19 RCTs [31–49] published between 2010 and 2023 were included, all of which were conducted in China. A total of 1921 patients (961 in intervention group and 960 in control group) were involved, with sample sizes ranging from 24 to 240 patients per trial. Twelve trials [32, 35, 36, 38–41, 43, 44, 46, 48, 49] reported the course of the disease, which ranged from 0.74 to 156 months. The age of patients varied greatly, ranging from 3 to 66 years old. The treatment duration was 2–48 weeks. Types of viral infectious diseases included Viral Myocarditis (VMC) (n = 7) [35, 37, 39–41, 44, 48], Hepatitis B (n = 6) [33, 34, 38, 43, 46, 49], Acquired Immune Deficiency Syndrome (AIDS) (n = 2) [45, 47], COVID-19 (n = 3) [31, 32, 42] and Hepatitis C (n = 1) [36]. Types of TCM syndrome differentiation included *Qi* and *Yin* deficiency (n = 9) [35, 37, 39–42, 44, 46, 48], *Qi* deficiency (n = 7) [31–34, 36, 43, 47], *Yin* deficiency (n = 1) [49], *Yang* deficiency (n = 1) [45] and spleen deficiency (n = 1) [38].

There were four categories of comparison in this review, including CHM plus drugs *vs*. drugs (n = 12) [33, 35, 36, 38–41, 43–47], CHM *vs*. drugs (n = 4) [34, 37, 48, 49], CHM *vs*. no treatment (n = 1) [42], CHM plus rehabilitation treatments *vs*. placebo plus rehabilitation treatments (n = 1) [32] and CHM *vs*. placebo (n = 1) [31]. The form of CHM included decoction (n = 8) [34–36, 38, 40, 41, 43, 48], granule (n = 4) [42, 44–46], capsule (n = 3) [32, 33, 37], oral liquid (n = 3) [31, 47, 49] and injection (n = 1) [39], which were taken 1–3 times daily. Eight of the trials used herbal patent, including pharmaceutical company production and hospital preparations. Fatigue as the primary outcome was measured by TCM syndrome scale (n = 17) [33–49], Fatigue Assessment Inventory (FAI) scores (n = 1) [32] and visual analogue scale (VAS) (n = 1) [31], higher scores indicated higher levels of fatigue. Secondary outcomes mainly included CD4 T lymphocytes (n = 6) [38, 39, 42, 43, 45, 47], CD8 T lymphocytes (n = 5) [38, 39, 42, 43, 45], IL-6 (n = 4) [35, 39, 41, 43] and TNF-α (n = 2) [35, 39]. Adverse events were reported in four of the studies [31, 32, 41, 46]. The characteristics of the included trials are summarized in Table 1, and the compositions of CHM formula are shown in Table 2.

## Risk of bias in studies

The risk of bias of the included trials was presented in Fig 2. Most of the involved studies had large methodological deficiencies, only 1 study [32] was assessed as having low risk of bias and 1 study [31] was assessed as having some concerns, while the remaining trials were all evaluated as having high risk of bias (Fig 2).

Forteen included studies [31–33, 35, 38–42, 45–49] described specific methods for random sequence generation, which used random number table, but only one study [32] reported using opaque envelopes for concealment of allocation. Three trials [31, 32, 45] reported the number and reason of dropouts, while no patients were reported withdrawing from the remaining 16 trials. Only 1 study [32] reported the estimation of sample size and 2 study reported the implementation of blinding [31, 32]. Fifteen studies [31–35, 38–42, 44–46, 48, 49] reported comparability of baseline data between groups.

## Effectiveness of interventions

To explore the source of the heterogeneity among trials, we tried subgroup analysis according to the protocol, however, none of them can lower the $I^2$ value within groups. Thus, we did not report neither the results of meta-analysis when $I^2 > 75\%$ nor the results of subgroup analysis. Funnel plot was not used since no meta-analysis with more than 10 included trials was conducted. Results of the included individual trials were shown in Tables 3 and 4 regarding to different outcomes.

## Primary outcome

**Fatigue scores.** One study [42] compared CHM to no treatment, which showed Yangyin Yiqi granules may improve the long-term fatigue after recovery of COVID-19 (MD = -0.80 scores, 95%CI -1.43 to -0.17 scores, *P* = 0.01, 60 participants, 1 trial) (Table 3).

One study compared CHM to placebo [31], which showed that Ludangshen oral liquid had significantly lower symptom score in fatigue for FAS analysis after SARS-CoV-2 infection (MD = -1.90 scores, 95%CI -2.38 to -1.42 scores, *P<0.00001*, 184 participants, 1 trial) (Table 3).

Another study [32] compared CHM to placebo on basis of rehabilitation therapy. The results showed that Bufei Huoxue capsule may be superior to placebo on relieving long-term fatigue measured by FAS scale after SARS-CoV-2 infection (MD = -14.90 scores, 95%CI -24.53 to -5.27 scores, *P* = 0.02, 118 participants, 1 trial) (Table 3).

**Table 1. Characteristics of the 18 included trials.**

| Study ID | Mean age(yrs) | | Sample size | Men (%) | Type of Disease | Disease duration(m) | | Intervention | Control | Outcomes | Treatment duration (w) |
|---|---|---|---|---|---|---|---|---|---|---|---|
| | T | C | | | | T | C | | | | |
| An XD 2023 | ≤30 9; 31–40 16; 41–50 23; 51–60 25; ≥61 26 | ≤30 5; 31–40 23; 41–50 21; 51–60 18;≥61 31 | 99/98 | 30.8 | COVID-19 | NR | | Ludangshen oral liquid | placebo of Ludangshen oral liquid | ⑪ | 2 |
| Chen YQ 2022 | 54.16 ±12.11 | 52.51 ±12.31 | 64/65 | 46.5 | COVID-19 | 3.14 ±0.57 | 3.14 ±0.68 | Bufei huoxue Capsules + rehabilitation therapy | Placebo capsules + rehabilitation therapy | ⑩ | 12 |
| Cui GT 2020 | 40.74 ±6.30 | 40.07±8.31 | 27/30 | 77.2 | Hepatitis B | NR | | Shenxi Capsule plus control | TDF, 300mg, once daily | ① | 24 |
| Guo ZJ 2012 | 38.44 ±9.06 | 39.28±8.70 | 120/120 | 60.8 | Hepatitis B | NR | | Qizhu Decoction | LAM, 1 capsule, once daily | ⑩ | 24 |
| Liang WZ 2019 | 45.37 ±4.87 | 45.37±4.17 | 60/60 | 65.8 | Hepatitis C | 42.84 ±22.44 | 42.84 ±14.88 | Xijiao Dihuang Decoction plus control | RBV (300mg, once weekly) + INF (180μg) | ⑩ | 48 |
| Li L 2021 | 5.37±1.15 | 5.34±1.11 | 33/32 | 76.9 | VMC | 0.74 ±0.21 | 0.75 ±0.21 | Gualou Huangqi Guizhi Decoction plus control | VitC (2g, once daily) + Creatinine (200mg, once daily) + 5% glucose injection (500ml, once daily) + RBV + IFN | ①⑥⑨ | 4 |
| Liu HJ 2010 | NR | | 34/34 | NR | VMC | NR | | Shensong Yangxin Capsule | RBV (0.1g, thrice daily) + Coenzyme Q10 (10mg, thrice daily) + VitC (0.1g, thrice daily) | ① | 4 |
| Ma SP 2015 | 37.20 ±9.30 | 38.70±8.70 | 79/75 | 50.6 | Hepatitis B | 36–144 | 24–156 | Jianpi Qinghua Decoction plus control | Recombinant Human Interferon α-2b for Injection, 5 million U bi-daily | ①②③④ | 24 |
| Niu FQ 2019 | 38.07 ±5.39 | 37.42±5.27 | 68/67 | 46.7 | VMC | 2.78 ±1.08 | 2.74 ±1.03 | Astragalus Injection plus control | VitC + Coenzyme Q10 + Coenzyme A + Taurine granules (1.6g, thrice daily) | ①②③④⑥⑨ | 4 |
| Peng J 2017 | 29.37 ±11.69 | 33.60 ±14.21 | 49/49 | 42.9 | VMC | 56.39 ±17.27 | 53.48 ±16.90 | Huangqi Zhenzhumu Decoction plus control | VitC + Fructose diphosphate injection + SiJi KangBingDu HeJi + Naloxone (1.2mg, twice daily) | ① | 2 |
| Qu WJ 2020 | 7.72±1.93 | 7.68±1.89 | 41/41 | 57.3 | VMC | 14.12 ±5.76 | 14.43 ±4.68 | Yinxin Decoction plus control | VitC (two tablets, thrice daily) + Coenzyme A plus 5% glucose solution (50-200U, twice daily) + Creatine Phosphate Sodium (0.5-1g) | ①⑥⑧ | 8 |
| Shi SF 2020 | 50.93 ±10.83 | 50.23 ±10.67 | 30/30 | 36.7 | COVID-19 | NR | | Yangyin Yiqi Granules | No treatment | ①②③④ | 2 |
| Tao L 2017 | 38.20 ±10.70 | 35.60±8.60 | 55/55 | 60 | Hepatitis B | 6–132 | 6–156 | Modified-Xiayuxue Decoction plus control | ETV, 0.5mg once daily | ①②③④⑤⑥⑦ | 24 |
| Wang Y 2017 | 5.80±2.11 | 5.94±2.30 | 30/30 | 51.7 | VMC | 4.96 ±1.47 | 4.98 ±1.60 | Shuangshen Yixin Decoction plus control | VitC (100mg, thrice daily) + Coenzyme Q10 (10mg, thrice daily) + Fructose diphosphate oral liquid (10ml, thrice daily) | ① | 4 |

*(Continued)*

**Table 1.** (Continued)

| Study ID | Mean age(yrs) | | Sample size | Men (%) | Type of Disease | Disease duration(m) | | Intervention | Control | Outcomes | Treatment duration (w) |
|---|---|---|---|---|---|---|---|---|---|---|---|
| | T | C | | | | T | C | | | | |
| Xiao TY 2021 | 57.61±6.45 | | 32/34 | 77.3 | AIDS | NR | | Wenyang Jianpi Huashi Decoction plus control | HAART | ①②③④ | 24 |
| Xuan CF 2018 | 26–61 | 26–61 | (12/12) | NR | AIDS | NR | | Compound Fufangteng Mixture plus control | HAART | ①② | 24 |
| Xu WD 2022 | 61.12 ±3.12 | 60.99±3.14 | 40/40 | 60 | Hepatitis B | 99.00 ±24.24 | 97.80 ±24.12 | Ganshuang Granules plus control | ETV, 0.5mg once daily | ① | 12 |
| Zhang C 2014 | 29.52 ±19.34 | 25.57 ±17.72 | 55/55 | 38.2 | VMC | 6–11.5 | 6.5–12 | Self-prescript herbal decoction | Glucose-insulin-potassium solution (once daily) + coenzyme Q10 (10mg, thrice daily) + VitC (0.2g, thrice daily) + VitE (20mg, thrice daily) | ① | 4 |
| Zhang QE 2014 | 50.71 ±10.27 | 50.69 ±10.97 | 40/40 | 48.8 | Hepatitis B | 18.39 ±2.07 | 18.17 ±2.21 | Rouganjiang-enzyme Mixture | Diammonium Glycyrrhizinate Enteric-coated Capsules, 3 capsules (50mg), thrice daily | ① | 12 |

T: Treatment group, C: Control group; m: months, w: weeks; NR: not report; AIDS: Acquired Immune Deficiency Syndrome; VMC: Viral Myocarditis; COVID-19: Corona Virus Disease 2019; ETV: Entecavir; TDF: Tenofovir Disoproxil Fumarate; LAM: Lamivudine; RBV: Ribavirin; INF: Interferon; HAART: Highly Active Antiretroviral Therapy; Vit: vitamin

Outcomes:①TCM syndrome score ②CD4+ ③CD8+ ④CD4+/CD8+ ⑤IL–2 ⑥IL–6 ⑦IL–10 ⑧IL–17 ⑨TNF-α ⑩FAI ⑪VAS

Four studies [34, 37, 48, 49] compared CHM to drugs, which involving 498 patients and two viral infectious diseases (viral hepatitis and viral myocarditis). The results of meta-analysis showed that CHM was better on improving fatigue symptoms (MD = -0.38 scores, 95%CI -0.48 to -0.27 scores, $I^2$ = 0%, $P$<0.00001, 498 participants, 4 trials) (Fig 3).

Twelve trials [33, 35, 36, 38–41, 43–47] compared CHM plus drugs to drugs alone, involving 1050 patients and three viral infectious diseases (viral hepatitis, HIV and viral myocarditis). All the studies reported the fatigue scores assessed through TCM syndrome scale. Higher scores indicated higher levels of fatigue. The results showed that, although 2 studies [29, 35] found no difference between groups on increasing the fatigue scores, other ten trials all suggested that CHM combined with drugs had better effect than drugs alone in improving the fatigue symptom, however, due to the obvious heterogeneity among included trials, pooling analysis was not conducted for these twelve trials (Table 3).

## Secondary outcomes

**Immunological indicators.** *T lymphocytes.* We identified six RCTs [38, 39, 42, 43, 45, 47] reported the CD4 T lymphocytes levels and five RCTs [38, 39, 42, 43, 45] reported the CD8 T lymphocytes levels. However, due to the different reporting forms of the immunity index, which including the number of cells and percentage. Different units could not be uniformly analyzed. Therefore, only 4 studies [38, 39, 42, 43] and 3 studies [38, 39, 43] with the same units were selected for meta-analysis (Table 4).

One study [42] showed that there was no statistical difference between Yiqi-yangyin granules and no intervention in improving the percentage of CD8 T lymphocytes during COVID-

**Table 2. Components of the included Chinese herbal prescriptions.**

| Study | Prescriptions of the HM | Form of the HM | Manufacturer | Treatment principle | Components | Method of administration |
|---|---|---|---|---|---|---|
| An XD 2023 | Ludangshen oral liquid | Oral liquid | Shanxi Zhenglai Pharmaceutical Co., Ltd | Tonifying *Qi* and tonifying spleen and lung | Ludangshen | 10ml, twice daily |
| Chen YQ 2022 | Bufei huoxue Capsules | Capsule | Guangdong Leiyunshang Pharmaceutical Co., Ltd | Tonifying *Qi* and tonifying spleen and lung | Radix Astragali, Radix paeoniae rubra, Fructus Psoraleae | 4 capsules, thrice daily |
| Cui GT 2020 | Shenxi Capsule | Capsule | Chia Tai Tianqing Pharmaceutical Group Co., Ltd | Invigorating *Qi* and promoting blood circulation | Radix Astragali, Radix Salviae miltiorrhizae, Radix Acanthopanacis Semticosi, Acanthopanax senticosus | 1g, thrice daily |
| Guo ZJ 2012 | Qizhu Decoction | Decoction | - | Invigorating *Qi* and promoting blood, clearing heat and promoting diuresis | Radix Astragali 15g, Phyllanthus urinaria 30g, Rhizoma Smilacis Glabrae 30g, Rhizoma polygoni cuspidati 15g, Hedyotis diffusa 30g, Radix paeoniae rubra 10g, Radix Salviae miltiorrhizae 10g | 100ml, twice daily |
| Liang WZ 2019 | Xijiao Dihuang Decoction | Decoction | - | Invigorating *Qi* and promoting blood circulation | Cornu Bubali 15g, Codonopsis pilosula 15g, Rhizoma Atractylodis Macrocephalae 10g, Radix Glycyrrhizae 5g, Radix rehmanniae12g, Radix Ophiopogonis 10g, Fructus schisandrae 10g, Radix Glehniae 15g, Rhizoma Polygonati Odorati 12g, Radix paeoniae rubra 14g, Radix Salviae miltiorrhizae 10g, Hedyotis diffusa 35g, Sedum sarmentosum Bunge 35g, radix isatidis 20g, Radix Bupleuri 14g | 250ml, twice daily |
| Li L 2021 | Gualou Huangqi Guizhi Decoction | Decoction | - | Tonifying *Qi* and *Yin* | Fructus Trichosanthis 15g, Radix rehmanniae 15g, Bulbus Allii Macrostemonis 15g, Radix Astragali 12g, Radix Ginseng 12g, Radix Paeoniae Alba 10g, Radix Salviae miltiorrhizae 10g, Ramulus Cinnamomi 10g, Radix Glehniae 10g, Radix Ophiopogonis 10g, Fructus schisandrae 10g, Aconiti Radix Lateralis Praeparata 10g, Rhizoma Zingiberis 10g, Rhizoma Pinelliae 9g, Radix Glycyrrhizae Preparata 9g | thrice daily |
| Liu HJ 2010 | Shensong Yangxin Capsule | Capsule | Beijing Yiling Pharmaceutical Co., Ltd | Tonifying *Qi* and *Yin* | Radix Ginseng, Radix Ophiopogonis, Cornus officinalis, Radix Salviae miltiorrhizae, Semen Ziziphi Spinosae (fry), Herba Taxilli, Radix paeoniae rubra, Eupolyphaga seu Steleophaga, Nardostachys chinensis Batal, Rhizoma Coptidis, Kadsura longepedunculata, Os Draconis | 3 capsules, thrice daily |
| Ma SP 2015 | Jianpi Qinghua Decoction | Decoction | - | Invigorating spleen to eliminate dampness | Radix Bupleuri 12g, Radix Astragali 20g, Pericarpium Citri Reticulatae 15g, Semen Coicis 15g, Radix Clematidis 15g, Fructus chaenomelis 15g, Endoconcha Sepiae 15g, Hedyotis diffusa 30g, Herba Cistanches 15g | twice daily |
| Niu FQ 2019 | Astragalus Injection | Injection | Shineway Pharmaceutical Group Co., Ltd | Tonifying *Qi* and *Yin* | Radix Astragali | 10ml, once daily |
| Peng J 2017 | Huangqi Zhenzhumu Decoction | Decoction | - | Tonifying *Qi* and *Yin* | Radix Astragali 30g, Margaritifera concha 30g, Radix Pseudostellariae 20g, Radix Ophiopogonis 15g, Semen Ziziphi Spinosae (fry) 15g, Carthamus tinctorius 10g, Rhizoma Polygonati Odorati 10g, Radix paeoniae rubra 10g, Radix Curcumae 10g, Fructus Aurantii 10g, Radix Polygalae 10g, Panax notoginseng 2g | twice daily |

(*Continued*)

**Table 2.** (Continued)

| Study | Prescriptions of the HM | Form of the HM | Manufacturer | Treatment principle | Components | Method of administration |
|---|---|---|---|---|---|---|
| Qu WJ 2020 | Yinxin Decoction | Decoction | - | Tonifying *Qi* and *Yin* | Radix Astragali 50g, Radix Ophiopogonis 15g, Radix rehmanniae 15g, Radix Pseudostellariae 10g, Radix Salviae miltiorrhizae 10g, Semen Ziziphi Spinosae 10g, Rhizoma Chuanxiong 10g, Fructus schisandrae 10g, Radix Polygalae 10g, Radix Glycyrrhizae Preparata 5g | twice daily |
| Shi SF 2020 | Yangyin Yiqi Granules | Granules | - | Tonifying *Qi* and *Yin* | Codonopsis pilosula (fry), Radix Ophiopogonis, Lilii Bulbus, Poria cocos, Rhizoma Atractylodis Macrocephalae (fry), Pericarpium Citri Reticulatae, Fructus Hordei Germinatus, Cortex Albiziae, Cortex Lycii, Radix Glycyrrhizae Preparata | twice daily |
| Tao L 2017 | Modified-Xiayuxue Decoction | Decoction | - | Invigorating *Qi* and promoting blood circulation | Radix et Rhizoma Rhei 9g, Semen Persicae 9g, Eupolyphaga seu Steleophaga 9g, Radix Astragali 30g, Radix Glycyrrhizae Preparata 5g | 200ml, twice daily |
| Wang Y 2017 | Shuangshen Yixin Decoction | Granules | - | Tonifying *Qi* and *Yin* | Radix Pseudostellariae 6~10g, Radix rehmanniae 6~10g, Radix Ophiopogonis 6~10g, Fructus schisandrae 6~8g, Rhizoma Polygonati Odorati 6~10g, Dioscorea polystachya 6~8g, Poria cocos 6~10g, Radix Salviae miltiorrhizae 6~10g, Rhizoma Chuanxiong 6~8g, Fructus Forsythiae 6~8g, Albizzia julibrissin Durazz 6~10g, Radix Glycyrrhizae Preparata 3~6g | twice daily |
| Xiao TY 2021 | Wenyang Jianpi Huashi Decoction | Granules | - | Invigorating spleen and resolving dampness | cortex cinnamomi, Carapax et Plastrum Testudinis, Fructus Amomi, Codonopsis pilosula, Rhizoma Atractylodis Macrocephalae, Fructus citrus sarcodactylis, Poria cocos, Aconiti Radix Lateralis Praeparata, Rhizoma Zingiberi, Radix Glycyrrhizae Preparata, Radix Salviae miltiorrhizae, Radix Angelica sinensis, Os Draconis, Lycium chinense, Fructus Aurantii (fried with bran) | 100ml, thrice daily |
| Xuan CF 2018 | Compound Fufangteng Mixture | Oral liquid | Guangxi University of Traditional Chinese Medicine Pharmaceutical Factory | Invigorating *Qi* and promoting blood circulation | Euonymus fortunei, Radix Astragali, Radix Ginseng Rubra, saccharose | 15ml, twice daily |
| Xu WD 2022 | Ganshuang Granules | Granules | Baoding Tianhao Pharmaceutical Co., Ltd | Tonifying *Qi* and *Yin* | Codonopsis pilosula, Radix Bupleuri, Radix Paeoniae Alba, Radix Angelica sinensis, Poria cocos, Rhizoma Atractylodis Macrocephalae (fry), Fructus Aurantii, Taraxacum mongolicum, Rhizoma polygoni cuspidati, Spica Prunellae, Radix Salviae miltiorrhizae, Semen Persicae, Carapax Trionycis | 3g, thrice daily |
| Zhang C 2014 | Self-prescript herbal decoction | Decoction | - | Tonifying *Qi* and *Yin* | Codonopsis pilosula 10g, Radix Astragali 25g, Radix Salviae miltiorrhizae 20g, Radix Ophiopogonis 15g, Radix rehmanniae 15g, Semen Ziziphi Spinosae (fry) 30g, Platycladi Semen 20g, Fructus schisandrae 10g, Radix Glycyrrhizae Preparata 9g, Poria cocos 10g, Pericarpium Citri Reticulatae 15g, Radix Paeoniae Alba 20g, Rhizoma Corydalis 10g, Radix Sophorae Flavescentis 10g | twice daily |
| Zhang QE 2014 | Rouganjiang-enzyme Mixture | Oral liquid | Minda Hospital of Hubei Minzu University | Nourishing liver and kidney | Fructus schisandrae, Radix Astragali, Fructus Crataegi, Fructus Jujubae | 30ml, thrice daily |

HM: Herbal medicine

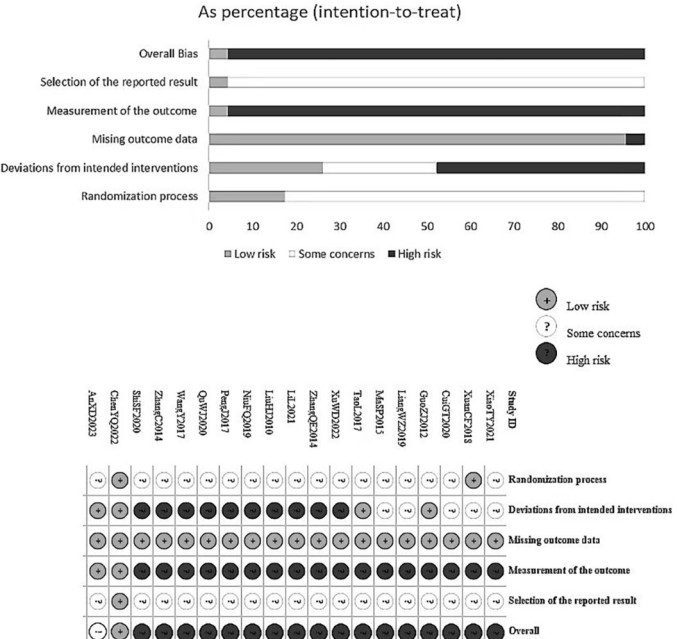

**Fig 2. Evaluation of methodological quality of the included trials through Cochrane risk.**

19 recovery (MD = 1.11 scores, 95%CI -2.91 to 5.13 scores, *P* = 0.59, 60 participants, 1 trial) (Table 4).

Three studies [38, 39, 43] compared CHM plus drugs to drugs alone, which involving 399 patients and two viral infectious dieases (hepatitis B and viral myocarditis), reported both CD4 and CD8 T lymphocytes levels. The results showed that in terms of CD4 T lymphocytes levels, CHM combined antiviral therapy had better efficacy than antiviral therapy in increasing the percentage of CD4 T lymphocytes. However, there was no statistical difference in the reduction of CD8 T lymphocytes levels (Table 4).

## TNF-α

Serum TNF-α levels was reported in two studies [35, 39]. The results showed that there was no statistical difference in reducing TNF-α comparing CHM plus drugs to drugs (Table 4).

## IL-6

Four studies [35, 39, 41, 43] reported serum IL-6 levels, but only two of them [26, 32] were used for data analysis to ensure unit consistency. The results showed that compared with drugs, CHM combined with drugs significantly reduced the level of serum IL-6 (MD = -14.64 scores, 95%CI 18.36 to -10.91 scores, $I^2$ = 0%, *P*<0.00001, 146 participants, 2 trials) (Fig 4).

## Adverse events

Adverse effects were reported in four studies [31, 32, 41, 46], mainly including diarrhea, nausea and vomiting, dizzy, somnolence, anemia, abnormal liver function, liver injury, excessive menstruation, nose bleeding and headache (see Table 5 for details). The remaining 15 trials did not report any adverse reactions.

**Table 3. Effect estimates of Chinese herbal medicine for post-viral fatigue concerning fatigue scores in 18 included trials.**

| Study ID | Type of Disease | Sample size | Effect Estimate (MD [95%CI]) | P-value |
|---|---|---|---|---|
| **CHM vs. No treatment** | | | | |
| Shi SF 2020 | COVID-19 | 60 | -0.80 [-1.43, -0.17] | 0.01 |
| **CHM vs. placebo** | | | | |
| An XD 2023 | COVID-19 | 184 | -1.90 [-2.38, -1.42] | <0.00001 |
| **CHM plus rehabilitation treatments vs. placebo plus rehabilitation treatments** | | | | |
| Chen YQ 2022 | COVID-19 | 118 | -14.90 [-24.53, -5.27] | 0.002 |
| **CHM vs. drugs** | | | | |
| Guo ZJ 2012 | Hepatitis B | 240 | -0.35 [-0.51, -0.19] | <0.0001 |
| Liu HJ 2010 | VMC | 68 | -0.45 [-0.71, -0.19] | 0.0008 |
| Zhang C 2014 | VMC | 110 | -0.34 [-0.52, -0.16] | 0.0002 |
| Zhang QE 2014 | Hepatitis B | 80 | -0.50 [-0.83, -0.17] | 0.003 |
| **CHM plus drugs vs. drugs** | | | | |
| Cui GT 2020 | Hepatitis B | 57 | -0.59 [-1.09, -0.09] | 0.02 |
| Liang WZ 2019 | Hepatitis C | 120 | -1.14 [-1.20, -1.08] | <0.00001 |
| Li L 2021 | VMC | 64 | -0.55 [-0.58, -0.52] | <0.00001 |
| Ma SP 2015 | Hepatitis B | 154 | -0.04 [-0.24, 0.16] | 0.7 |
| Niu FQ 2019 | VMC | 135 | -0.68 [-0.89, -0.47] | <0.00001 |
| Peng J 2017 | VMC | 98 | -0.44 [-0.63, -0.25] | <0.00001 |
| Qu WJ 2020 | VMC | 82 | -0.23 [-0.43, -0.03] | 0.03 |
| Tao L 2017 | Hepatitis B | 110 | -0.68 [-0.88, -0.48] | <0.00001 |
| Wang Y 2017 | VMC | 60 | -0.14 [-0.51, 0.23] | 0.46 |
| Xiao TY 2021 | AIDS | 66 | -1.73 [-2.19, -1.27] | <0.00001 |
| Xuan CF 2018 | AIDS | 24 | -0.84 [-1.51, -0.17] | 0.01 |
| Xu WD 2022 | Hepatitis B | 80 | -0.36 [-0.44, -0.28] | <0.00001 |

MD: Mean Difference, CI: Confidence Interval, VMC: Viral Myocarditis, AIDS: Acquired Immune Deficiency Syndrome

## GRADE assessment

Table 6 shows a summary of the overall quality of evidence assessment for the effect of CHM on the fatigue scores in different comparison. Certainty in the evidence was variable for CHM vs. no treatment (Very low), CHM vs. Placebo (Low), CHM vs. placebo on basis of rehabilitation therapy (Low), CHM vs. drugs (Very low), CHM plus drugs vs. drugs (Very low). All evidence were downgraded due to imprecision, other reasons for downgrading included high risk of the bias, obvious heterogeneity and probability of the publication bias (Table 6).

## Discussion

### Summary of the main findings

This review found that CHM has good effect as adjuvant therapy or monotherapy for viral infectious diseases on improving post-viral fatigue. CHM may ameliorate patients' fatigue symptom, with an average lower of 0.56 points in TCM fatigue syndrome scale (CHM vs. Drugs 0.38 scores lower; CHM plus drugs vs. Drugs 0.50 scores lower; CHM vs. no treatment 0.8 scores lower), and enhance body immunity, with an average increase of 5.26% in CD4 T lymphocytes percentage and lower of 14.64ng/L in serum IL-6 level.

**Table 4. Effect estimates of Chinese herbal medicine for post-viral fatigue concerning T lymphocytes and TNF-α in included trial.**

| Study ID | Type of Disease | Sample size | Effect Estimate (MD [95%CI]) | P-value |
|---|---|---|---|---|
| **CD4 T lymphocytes** | | | | |
| **CHM *vs*. No treatment** | | | | |
| Shi SF 2020 | COVID-19 | 60 | 1.11 [-2.91, 5.13] | 0.59 |
| **CHM plus drugs *vs*. drugs** | | | | |
| Ma SP 2015 | Hepatitis B | 154 | 4.39 [3.27, 5.51] | <0.00001 |
| Niu FQ 2019 | VMC | 135 | 10.41 [8.28, 12.54] | <0.00001 |
| Tao L 2017 | Hepatitis B | 110 | 1.08 [-0.91, 3.07] | 0.29 |
| **CD8 T lymphocytes** | | | | |
| **CHM plus drugs *vs*. drugs** | | | | |
| Ma SP 2015 | Hepatitis B | 154 | -2.82 [-4.43, -1.21] | 0.0006 |
| Niu FQ 2019 | VMC | 135 | 0.11 [-1.67, 1.89] | 0.9 |
| Tao L 2017 | Hepatitis B | 110 | -7.08 [-8.48, -5.68] | <0.00001 |
| **TNF-α** | | | | |
| **CHM plus drugs *vs*. drugs** | | | | |
| Li L 2021 | VMC | 65 | -7.10 [-8.91, -5.29] | <0.00001 |
| Niu FQ 2019 | VMC | 135 | -1.29 [-1.49, -1.09] | <0.00001 |

MD: Mean Difference, CI: Confidence Interval, VMC: Viral Myocarditis, AIDS: Acquired Immune Deficiency Syndrome

However, there was significant clinical heterogeneity among the included trials in this review. We had conducted subgroup meta-analysis according to the different types of virus infection, different disease stages after virus infection, the different TCM treatment principle (e.g. replenishing qi/blood/yin/yang), and different treatment duration, but none of them could explain the source of heterogeneity. This indicated that the heterogeneity may induce by complex factors. Therefore, the evidence we provided was only of a small amount of CHM with tonifying property to post-viral fatigue, and the reliability of the results needed further research and discussion.

Based on the existing TCM syndrome score, a total of 15 trials significantly reduced fatigue compared to the control group, with a maximum lower of 1.73 scores and an average lower of 0.59 scores (CHM vs. Drugs 0.38 scores lower; CHM plus drugs vs. Drugs 0.58 scores lower; CHM vs. no treatment 0.8 scores lower). There was no significant difference in two trials compared with control group, with a minimum lower of 0.04 scores, and the disease types were chronic hepatitis B and viral myocarditis.

In addition, only four studies reported the adverse events, one of which [32] resulted in abnormal liver function. Although there was no difference concerning the incidence of adverse events between groups, safety of CHM in treating post-viral fatigue was still uncertain due to the insufficient evidence.

## Limitations

Most of the included trials had methodological limitations, 26.32% of the trials failed to provide the methods for random sequence generation, only two trials described the blinding and one of them used [32] concealment of allocation. Therefore, all of these methodological deficiencies lead to bias.

At the same time, as mentioned above, five viral infectious diseases and 18 CHM preparations with different compositions were included in this review, there was great clinical

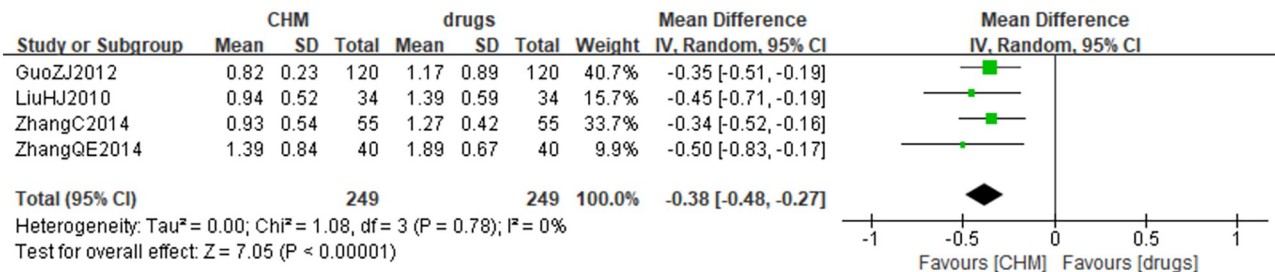

**Fig 3. Forest plot of Chinese herbal medicine versus drugs on improving post-viral fatigue according to TCM syndrome scale.**

heterogeneity among the trials. We included 5 kinds of virus infectious diseases. According to the route of transmission, the virus could be grouped into respiratory tract transmission virus, digestive tract transmission virus, contact transmission virus and blood body fluid transmission virus. And according to the status of virus infection, it also could be divided into virus negative but the body still having persistent fatigue (e.g. Long COVID, chronic viral myocarditis) and virus positive, fatigue and virus having a certain correlation (e.g. chronic hepatitis B, AIDS). Therefore, viruses invaded the human body in different ways, different infection states, resulting in high heterogeneity among studies.

In terms of interventions, modern medicine mainly used antiviral drugs in the treatment of virus-related fatigue. However, TCM was characterized by syndrome differentiation and treatment. Therefore on the basis of the treatment principles of supplementing qi, blood, yin, and/ or yang, we usually added or subtraction on basic prescriptions depending on the patients, which leaded to the diversification of prescriptions. It was also the reason for the large heterogeneity among clinical trials.

In addition, the main outcome measurement tool used in this review was the TCM syndrome scale, which might cause bias due to the subjective evaluation criteria. Only two included trials used internationally recognized fatigue scale, most of the remaining trials used TCM syndrome score (which contained scores for fatigue) to assess the improvement of clinical symptoms. TCM syndrome score is a commonly used scale in China, which is guided by TCM syndrome differentiation to evaluate the efficacy of TCM in improving symptoms and signs. In this study, we used the score for evaluating fatigue level in the scale as the primary outcome. Its advantage is that it is highly relevant to the diagnosis of TCM. But we have to admit that this scale lacks a unified evaluation standard, which can not measure the fatigue from multiple dimensions as a whole. We suggest that future researchers try to use internationally recognized outcome measurements as much as possible.

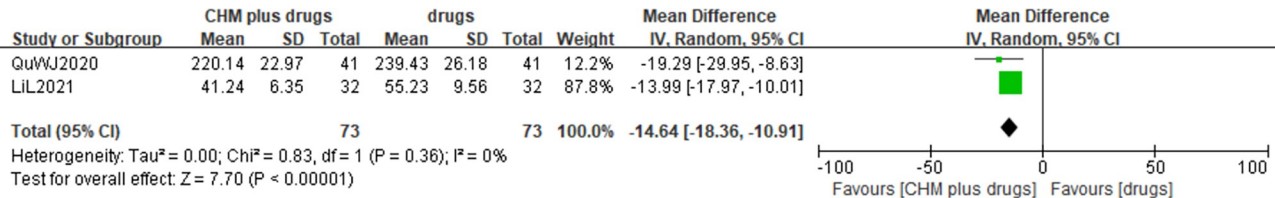

**Fig 4. Forest plot of Chinese herbal medicine combined with drugs versus drugs on improving the level of serum IL-6.**

**Table 5. Number of the adverse events in included trials.**

| Study ID | Adverse events (No. of cases) | |
|---|---|---|
| | **Treatment group** | **Control group** |
| An XD 2023 | Nose bleeding (1) | Headache (1) |
| Chen YQ 2022 | Abnormal liver function (4), Liver injury (1), Diarrhea (1) | Abnormal liver function (2), Excessive menstruation (1) |
| Qu WJ 2020 | Dizzy (1), Nausea and vomiting (1), Somnolence (2), Anemia (1) | Dizzy (1), Nausea and vomiting (3), Somnolence (1), Anemia (1) |
| Xu WD 2022 | Diarrhea (1), Nausea and vomiting (1), Dizzy (1) | Allergy (1), Diarrhea (1), Nausea and vomiting (1), Dizzy (1) |

**Table 6. GRADE quality of evidence assessment for fatigue scores in different comparison.**

Patient or population: post-viral fatigue
Setting: in hospital

| Outcomes | Anticipated absolute effects* (95% CI) | | № of participants (studies) | Certainty of the evidence (GRADE) |
|---|---|---|---|---|
| | **Risk with control** | **Risk with experimental** | | |
| CHM *vs.* no treatment | The mean CHM vs no treatment was **1.93 scores** | MD **0.8 scores lower** (1.43 lower to 0.17 lower) | 60 (1 RCT) | ⊕○○○ Very low [a,b] |
| CHM *vs.* placebo | The mean CHM vs no treatment was **3.21 scores** | MD **1.90 scores lower** (2.38 lower to 1.42 lower) | 197 (1 RCT) | ⊕⊕○○ Low [a,b] |
| CHM *vs.* placebo on basis of rehabilitation therapy | The mean CHM vs placebo was **100.4 scores** | MD **14.9 scores lower** (24.53 lower to 5.27 lower) | 118 (1 RCT) | ⊕⊕○○ Low [b] |
| CHM *vs.* drugs | The mean CHM vs drugs was **1.34 scores** | MD **0.38 scores lower** (0.48 lower to 0.27 lower) | 498 (4 RCTs) | ⊕○○○ Very low [a,b,c] |
| CHM plus drugs *vs.* drugs | The mean CHM plus drugs vs drugs was **1.35 scores** | MD **0.59 scores lower** (0.81 lower to 0.38 lower) | 1050 (12 RCTs) | ⊕○○○ Very low [a,c,d] |

*__The risk in the intervention group__ (and its 95% confidence interval) is based on the assumed risk in the comparison group and the **relative effect** of the intervention (and its 95% CI).

**CI**: confidence interval; **MD**: mean difference

GRADE Working Group grades of evidence

**High certainty**: we are very confident that the true effect lies close to that of the estimate of the effect.

**Moderate certainty**: we are moderately confident in the effect estimate: the true effect is likely to be close to the estimate of the effect, but there is a possibility that it is substantially different.

**Low certainty**: our confidence in the effect estimate is limited: the true effect may be substantially different from the estimate of the effect.

**Very low certainty**: we have very little confidence in the effect estimate: the true effect is likely to be substantially different from the estimate of effect.

**Explanations**

[a]. Lack of allocation concealment and blinding method indicate high risk of selection and detection bias

[b]. Wide confidence intervals and small sample size indicate less imprecision of the results

[c]. Large $I^2$ value indicate obvious heterogeneity among included trials

[d]. High probability of publication bias

## Implications for clinical practice

Subgroup meta-analysis did not reveal any significant influence of factors such as syndrome differentiation, types of viral infection and duration of treatment in the category of CHM *vs.* drugs.

This review found that Radix Astragalus 10~50g, Radix Salvia miltiorrhiza 6~20g, Radix Ophiopogon 6~15g and Poria cocos 6~10g were the most frequently used herbs for post-viral fatigue. Modern pharmacological research showed that Astragalus promoted the development of immune organs, enhanced mucosal immune function, increased the quantity and phagocytic capacity of innate immunity, promoted the maturation and differentiation of acquired immunity cells, and improves the expression of antibodies in acquired immunity [50]. Tanshinone IIA could improve microcirculation and participate in the development and activation of immune cells [51]. In TCM theory, Astragalus and Salvia miltiorrhiza nourished the human body from the perspective of *Qi* and blood respectively to improve chronic fatigue. Through long-term clinical verification, the efficacy had been widely recognized. Ophiopogonin D and carboxymethyl-pachyman through enhanced exercise endurance, reduced lactic acid accumulation and increased glycogen reserve in mice to produce great anti-fatigue effect [52, 53].

## Implications for future research

Chronic fatigue is a prominent symptom after virus infection, causing psychological and physical harm to patients, seriously interferes with people's life. However, in a review of 19 studies, we found that most of the existing studies only evaluated fatigue as one of the many symptoms after viral infection using the TCM syndrome scale. The subjectivity of evaluation is strong, lacking of objective quantitative analysis of fatigue. Therefore, future clinical trials should focus on the symptom of fatigue and use authoritative fatigue measurement tools (e.g. Chalder fatigue scale [27]) to evaluate. This will contribute to explore insight into the effectiveness of CHM in improving post-viral fatigue.

In addition, this review is an overall evaluation of the efficacy of all existing CHM with tonifying property in the treatment of post-viral fatigue, but limited by the number of existing trials, no reliable results have been obtained. However, taking this study as an opportunity, there are still many ideas that can be further explored. It includes a individual evaluation of the efficacy of CHM for symptoms caused by a specific type of virus (e.g.respiratory viruses, gastrointestinal viruses.etc) and a separately evaluation of CHM for fatigue in different virus infection status (e.g. virus continued positive or has turned negative). Therefore, with the increase of the number of trials in the future, in-depth research can be carried out for a certain direction.

At the same time, from the perspective of methodology, it is also necessary for researchers to carry out more high-quality RCTs with multiple centers and large samples. In the research process, the random sequence generation, the concealment of allocation and blinding should be strictly implemented, and the shedding situation of patients, adverse events and economic costs should be definite clarified. In addition, based on the recurring nature of chronic fatigue, future studies with longer follow-up times should be conducted to confirm the long-term efficacy and safety of CHM for post-viral fatigue.

## Conclusion

Current very low quality evidence showed that the participation of CHM can improve the symptoms of post-viral fatigue and some immune indicators. However, the safety of this complementary alternative therapy remains unknown. Large sample, high quality multi-center randomized controlled trials are needed to be conducted in the future.

## Supporting information

**S1 Appendix. PRISMA 2020 checklist.**
(DOCX)

**S2 Appendix. Search strategies.**
(DOCX)

## Author Contributions

**Conceptualization:** Hui-Juan Cao.

**Funding acquisition:** Zheng Li.

**Investigation:** Le-Yan Hu, An-Qi Cai.

**Methodology:** Le-Yan Hu, An-Qi Cai.

**Resources:** Bo Li.

**Supervision:** Jian-Ping Liu, Hui-Juan Cao.

**Writing – original draft:** Le-Yan Hu, Hui-Juan Cao.

**Writing – review & editing:** Le-Yan Hu, Hui-Juan Cao.

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
