## [Decision Letter · Decision Letter 0]

26 Oct 2023

PONE-D-23-31247Chinese herbal medicine for post-viral fatigueChinese herbal medicine for post-viral fatigue: a systematic review of randomized controlled trialsPLOS ONE

Dear Dr. Cao,

Thank you for submitting your manuscript to PLOS ONE. After careful consideration, we feel that it has merit but does not fully meet PLOS ONE’s publication criteria as it currently stands. Therefore, we invite you to submit a revised version of the manuscript that addresses the points raised during the review process.

The topic of this article is in line with the scope of this journal. The author provides a review of the clinical use of Chinese herbal medicine in the treatment of viral post fatigue, which has reference value. However, the manuscript has not yet met the publication requirements. Please review the entire text, correct any errors, and revise according to the reviewer's suggestions. Please pay special attention to whether the clinical studies cited are within the scope of inclusion.

We look forward to receiving your revised manuscript.

Kind regards,

Hongxun Tao

Academic Editor

PLOS ONE

Journal Requirements:

2. Please amend either the title on the online submission form (via Edit Submission) or the title in the manuscript so that they are identical.

Additional Editor Comments :

The topic of this article is in line with the scope of this journal. The author provides a review of the clinical use of Chinese herbal medicine in the treatment of viral post fatigue, which has reference value. Please revise the manuscript according to the reviewer's suggestions.

Reviewers' comments:

Reviewer's Responses to Questions

**Comments to the Author**

1. Is the manuscript technically sound, and do the data support the conclusions?

Reviewer #1: Yes

Reviewer #2: Yes

2. Has the statistical analysis been performed appropriately and rigorously? 

Reviewer #1: Yes

Reviewer #2: Yes

3. Have the authors made all data underlying the findings in their manuscript fully available?

Reviewer #1: Yes

Reviewer #2: Yes

4. Is the manuscript presented in an intelligible fashion and written in standard English?

Reviewer #1: Yes

Reviewer #2: Yes

5. Review Comments to the Author

Reviewer #1: The author chose a very interesting clinical question, and conducted a systematic review using standard procedure with comprhensive literature retrieval and strict quality evaluation, which concluded that some Chinese medicine prescriptions might improve the fatigue after virus infection. Some design and quality concerns, problems in the field were discussed, some solutions were put forward. In general it's a useful study to the field of clinicians, researchers and patients. Yet there are some minor points for improvement.

1. The clinical heterogeneity brought by the virus types and interventions (whether the prescription suits the condition in the view of either TCM or western medicine points of view) are the most important factors affecting the scientific soundness of the conclusion, and the generalization of the conclusions, thus the limitations and further improvement measures should be fully discussed;

2. Can we discuss the influence of the credibility of the TCM fatigue syndrome in specific details rather than a word in principle: not widely recognized/accepted, better use the internationally recognized fatigue scale, etc. In case its composition (content) is scientific enough, we can do the reliability and validity test further, etc.; if there is scientifically incorrect, in what respect? what about the size of the influence on the research conclusion?

3. For literature search: some papers with keywords not including traditional Chinese medicine, Chinese patent (eg. just indicated to assess the efficacy of some formula on a virus infection...)? Can we find some more from the clinical trial registration platform or structured clinical evidence database for TCM (published by Tianjin and Guangdong), it seems easy to find trials including fatigue in the field of outcome?

Reviewer #2: This paper reports on a systematic review of randomized controlled trials about Chinese herbal medicine for post-viral fatigue . The systematic review of randomized controlled trials is interesting, but the authors should better underline, concretely, how the results are useful. The manuscript needs to be revised, such as addressing spelling errors, incomplete sentences, annotating references, etc. Because these problems greatly affect readability. Here are some specific comments that can help authors strengthen their papers:

Introduction

1.The authors did not describe the prevalence of post-viral fatigue at home and abroad.

2.The authors did not describe the epidemiology of Chinese herbal medicine for post-viral fatigue.

Eligibility criteria

1.The authors described the inclusion criteria but did not describe the exclusion criteria.

Data analysis

1.In line 181, the percentage of statistical heterogeneity analysis omissions.

Discussion

1.The discussion appears to repeat the results, and did not appear to add much value.

6. PLOS authors have the option to publish the peer review history of their article (what does this mean?). If published, this will include your full peer review and any attached files.

Reviewer #1: No

Reviewer #2: No

---

## [Author Response · Author response to Decision Letter 0]

4 Dec 2023

Response to Reviewer #1:

(Comment 1) Authors responsed 'The clinical heterogeneity brought by the virus types and interventions (whether the prescription suits the condition in the view of either TCM or western medicine points of view) are the most important factors affecting the scientific soundness of the conclusion, and the generalization of the conclusions, thus the limitations and further improvement measures should be fully discussed.'

Response: Thank you for the comment. We added two paragraph in Limitation part of the Discussion section to further interpret these two source of the clinical heterogeneity. Also in the Summary of the main findings part of the discussion, we considered the impact of heterogeneity when explaining the main findings and provided a more cautious and objective expression of the results.

Discussion>Summary of the main findings>2nd paragraph and 3rd paragraph: 

" However, there was significant clinical heterogeneity among the included trials in this review. We had conducted subgroup meta-analysis according to the different types of virus infection, different disease stages after virus infection, the different TCM treatment principle (e.g. replenishing qi/blood/yin/yang), and different treatment duration, but none of them could explain the source of heterogeneity. This indicated that the heterogeneity may induce by complex factors. Therefore, the evidence we provided was only of a small amount of CHM with tonifying property to post-viral fatigue, and the reliability of the results needed further research and discussion.

Based on the existing TCM syndrome score, a total of 15 trials significantly reduced fatigue compared to the control group, with a maximum lower of 1.73 scores and an average lower of 0.59 scores (CHM vs. Drugs 0.38 scores lower; CHM plus drugs vs. Drugs 0.58 scores lower; CHM vs. no treatment 0.8 scores lower). There was no significant difference in two trials compared with control group, with a minimum lower of 0.04 scores, and the disease types were chronic hepatitis B and viral myocarditis."

Discussion>Limitations>2nd paragraph and 3rd paragraph: 

" We included 5 kinds of virus infectious diseases. According to the route of transmission, the virus could be grouped into respiratory tract transmission virus, digestive tract transmission virus, contact transmission virus and blood body fluid transmission virus. And according to the status of virus infection, it also could be divided into virus negative but the body still having persistent fatigue (e.g. Long COVID, chronic viral myocarditis) and virus positive, fatigue and virus having a certain correlation (e.g. chronic hepatitis B, AIDS). Therefore, viruses invaded the human body in different ways, different infection states, resulting in high heterogeneity among studies. 

In terms of interventions, modern medicine mainly used antiviral drugs in the treatment of virus-related fatigue. However, TCM was characterized by syndrome differentiation and treatment. Therefore on the basis of the treatment principles of supplementing qi, blood, yin, and/or yang, we usually added or subtraction on basic prescriptions depending on the patients, which leaded to the diversification of prescriptions. It was also the reason for the large heterogeneity among clinical trials." 

(Comment 2) Authors responsed 'Can we discuss the influence of the credibility of the TCM fatigue syndrome in specific details rather than a word in principle: not widely recognized/accepted, better use the internationally recognized fatigue scale, etc. In case its composition (content) is scientific enough, we can do the reliability and validity test further, etc.; if there is scientifically incorrect, in what respect? what about the size of the influence on the research conclusion?'

Response: Thank you for the comment. We totally agree with you. During the literature searching and screening, we tried to find trials that using an internationally accepted fatigue scale as the primary outcome. However, due to the limited number of studies, we only find two included trials, which concerned Long COVID, used the FAI scores and VAS respectively. Thus, we only reported the results of these two trials as individual study. In meta-analysis, we pooled the results of studies with TCM fatigue syndrome. Though the TCM syndrome scale is not widely recognized/accepted, it is commonly used in China, and also assess the degree of fatigue by scores. We have provided further discussion of this issue in “Limitations” section. We do hope that with the publication of this systematic review, researchers in related fields can realize the importance of internationally recognized scales and produce more high-quality clinical trials in the future. It is also hoped that more trials with internationally recognized outcome will be included in our future update review.

Discussion>Limitations>4thparagraph: 

"Only two included trials used internationally recognized fatigue scale, most of the remaining trials used TCM syndrome score (which contained scores for fatigue) to assess the improvement of clinical symptoms. TCM syndrome score is a commonly used scale in China, which is guided by TCM syndrome differentiation to evaluate the efficacy of TCM in improving symptoms and signs. In this study, we used the score for evaluating fatigue level in the scale as the primary outcome. Its advantage is that it is highly relevant to the diagnosis of TCM. But we have to admit that this scale lacks a unified evaluation standard, which can not measure the fatigue from multiple dimensions as a whole. We suggest that future researchers try to use internationally recognized outcome measurements as much as possible." 

(Comment 3) Authors responsed 'For literature search: some papers with keywords not including traditional Chinese medicine, Chinese patent (eg. just indicated to assess the efficacy of some formula on a virus infection...)? Can we find some more from the clinical trial registration platform or structured clinical evidence database for TCM (published by Tianjin and Guangdong), it seems easy to find trials including fatigue in the field of outcome?'

Response: Thank you for the suggestion. We update the literature searching from six electronic databases and search three clinical trial registration platforms moreover. We've included one more trial related to Long COVID. Therefore, the relevant data covered in the manuscript is modified. The “Fig 1. Flow chart of study selection” has been changed as follows:

Results>1stparagraph: 

Fig 1. Flow chart of study selection

Response to Reviewer #2:

(Comment 1) Authors responsed 'The authors did not describe the prevalence of post-viral fatigue at home and abroad.'

Response: Thank you for the comment. We added epidemiological information on chronic hepatitis B and AIDS related fatigue. And in Introduction>2nd paragraph we mentioned the prevalence of Long COVID related fatigue.

Introduction>2nd paragraph:

" The results of AIDS and hepatitis B with large sample study in China show that the prevalence rate is up to 40% and even more than 60% in England and North America." 

Introduction>2nd paragraph:

" An meta-analysis have shown that at home and abroad approximately a third of individuals experienced persistent fatigue 12 or more weeks following confirmed COVID-19 diagnosis." 

(Comment 2) Authors responsed 'The authors did not describe the epidemiology of Chinese herbal medicine for post-viral fatigue.'

Response: Thank you for the comment. Taking Ginseng as an example, we described the anti-fatigue effect of Ginseng and its application in the treatment of Long COVID.

Introduction>5th paragraph:

" At present, although there is no direct evidence for the usage rate of CHM in the treatment of post-viral fatigue, but in the past few decades, it has been widely used in clinic and achieved certain results in the treatment of cancer-related fatigue (CRF) and chronic fatigue syndrome (CFS). Evidence suggests that CHM can clearly decreased 1.77 scores in fatigue scale-14 (FS-14) scores as an adjuvant or monotherapy for CFS and significantly lowered 1.47 scores in Brief Fatigue Inventory (BFI) global score for CRF. Meanwhile take Ginseng for example, up to now, various clinical practice and animal-based experiments have already confirmed the safe anti-fatigue effects of Ginseng Radix et Rhizoma, as well as its components. Ginseng Radix et Rhizoma is currently prescribed in the formula for Long COVID treatment as a monarch drug. Among these formulas, Qingjin Yiqi granules can significantly alleviate fatigue and have been recommended by the Rehabilitati on Guidelines of Integrated Medicine for Long COVID treatment in clinical." 

(Comment 3) Authors responsed 'The authors described the inclusion criteria but did not describe the exclusion criteria.'

Response: Thank you for pointing out this issue. We modified the main text to make it clear.

Eligibility criteria>3rdparagraph:

" Exclusion criteria：

1)Incomplete data or full text not available

2)Duplicate publication" 

(Comment 4) Authors responsed 'In line 181, the percentage of statistical heterogeneity analysis omissions.'

Response: Thank you for pointing out this error. We have modified and added the per cent (%).

Data analysis>1stparagraph:

" I2 <75%" 

(Comment 5) Authors responsed 'The discussion appears to repeat the results, and did not appear to add much value.'

Response: Thank you for the comment. In the discussion section, we further elaborated the limitations of this review, and specifically discussed the reasons for the increase of heterogeneity among trials caused by the type of viruses, intervention (increase or decrease of CHM prescription), and the primary outcome (TCM syndrome score). In implications for future research, we provided suggestions for future research directions.

Discussion>Summary of the main findings>2nd paragraph and 3rd paragraph: 

" However, there was significant clinical heterogeneity among the included trials in this review. We had conducted subgroup meta-analysis according to the different types of virus infection, different disease stages after virus infection, the different TCM treatment principle (e.g. replenishing qi/blood/yin/yang), and different treatment duration, but none of them could explain the source of heterogeneity. This indicated that the heterogeneity may induce by complex factors. Therefore, the evidence we provided was only of a small amount of CHM with tonifying property to post-viral fatigue, and the reliability of the results needed further research and discussion.

Based on the existing TCM syndrome score, a total of 15 trials significantly reduced fatigue compared to the control group, with a maximum lower of 1.73 scores and an average lower of 0.59 scores (CHM vs. Drugs 0.38 scores lower; CHM plus drugs vs. Drugs 0.58 scores lower; CHM vs. no treatment 0.8 scores lower). There was no significant difference in two trials compared with control group, with a minimum lower of 0.04 scores, and the disease types were chronic hepatitis B and viral myocarditis."

Discussion>Limitations>2nd paragraph and 3rd paragraph: 

" We included 5 kinds of virus infectious diseases. According to the route of transmission, the virus could be grouped into respiratory tract transmission virus, digestive tract transmission virus, contact transmission virus and blood body fluid transmission virus. And according to the status of virus infection, it also could be divided into virus negative but the body still having persistent fatigue (e.g. Long COVID, chronic viral myocarditis) and virus positive, fatigue and virus having a certain correlation (e.g. chronic hepatitis B, AIDS). Therefore, viruses invaded the human body in different ways, different infection states, resulting in high heterogeneity among studies. 

In terms of interventions, modern medicine mainly used antiviral drugs in the treatment of virus-related fatigue. However, TCM was characterized by syndrome differentiation and treatment. Therefore on the basis of the treatment principles of supplementing qi, blood, yin, and/or yang, we usually added or subtraction on basic prescriptions depending on the patients, which leaded to the diversification of prescriptions. It was also the reason for the large heterogeneity among clinical trials." 

Discussion>Limitations>4thparagraph: 

" Only two included trials used internationally recognized fatigue scale, most of the remaining trials used TCM syndrome score (which contained scores for fatigue) to assess the improvement of clinical symptoms. TCM syndrome score is a commonly used scale in China, which is guided by TCM syndrome differentiation to evaluate the efficacy of TCM in improving symptoms and signs. In this study, we used the score for evaluating fatigue level in the scale as the primary outcome. Its advantage is that it is highly relevant to the diagnosis of TCM. But we have to admit that this scale lacks a unified evaluation standard, which can not measure the fatigue from multiple dimensions as a whole. We suggest that future researchers try to use internationally recognized outcome measurements as much as possible." 

Discussion>Implications for future research>2ndparagraph:

" In addition, this review is an overall evaluation of the efficacy of all existing tonic CHM in the treatment of post-viral fatigue, but limited by the number of existing trials, no reliable results have been obtained. However, taking this study as an opportunity, there are still many ideas that can be further explored. It includes a individual evaluation of the efficacy of CHM for symptoms caused by a specific type of virus (e.g.respiratory viruses, gastrointestinal viruses .etc) and a separately evaluation of CHM for fatigue in different virus infection status (e.g. virus continued positive or has turned negative). Therefore, with the increase of the number of trials in the future, in-depth research can be carried out for a certain direction."

---

## [Decision Letter · Decision Letter 1]

6 Mar 2024

Chinese herbal medicine for post-viral fatigue: a systematic review of randomized controlled trials

PONE-D-23-31247R1

Dear Dr. Cao,

We’re pleased to inform you that your manuscript has been judged scientifically suitable for publication and will be formally accepted for publication once it meets all outstanding technical requirements.

Kind regards,

Timothy Omara, PhD

Academic Editor

PLOS ONE

Additional Editor Comments (optional):

Reviewers' comments:

Reviewer's Responses to Questions

**Comments to the Author**

1. If the authors have adequately addressed your comments raised in a previous round of review and you feel that this manuscript is now acceptable for publication, you may indicate that here to bypass the “Comments to the Author” section, enter your conflict of interest statement in the “Confidential to Editor” section, and submit your "Accept" recommendation.

Reviewer #3: All comments have been addressed

2. Is the manuscript technically sound, and do the data support the conclusions?

Reviewer #3: Yes

3. Has the statistical analysis been performed appropriately and rigorously? 

Reviewer #3: Yes

4. Have the authors made all data underlying the findings in their manuscript fully available?

Reviewer #3: Yes

5. Is the manuscript presented in an intelligible fashion and written in standard English?

Reviewer #3: Yes

6. Review Comments to the Author

Reviewer #3: This meta-analysis provided the latest evidnece of Chinese herbal medicine for post-viral fatigue on RCTs.

7. PLOS authors have the option to publish the peer review history of their article (what does this mean?). If published, this will include your full peer review and any attached files.

Reviewer #3: **Yes: **Yongliang Jia
